# The Health Consequences of Neocolonialism for Latin American Immigrant Women Working as Caregivers in Spain: A Multisite Qualitative Analysis

**DOI:** 10.3390/ijerph17218278

**Published:** 2020-11-09

**Authors:** Erica Briones-Vozmediano, Natalia Rivas-Quarneti, Montserrat Gea-Sánchez, Andreu Bover-Bover, Maria Antonia Carbonero, Denise Gastaldo

**Affiliations:** 1GESEC Group, Department of Nursing and Physiotherapy, University of Lleida, 25198 Lleida, Spain; erica.briones@udl.cat; 2Grup de Recerca en Cures de la Salut, Institut de Recerca Biomèdica de Lleida, 25198 Lleida, Spain; 3Global Migration and Health Initiative, Toronto, ON M5T 1P8, Canada; natalia.rivas.quarneti@udc.es (N.R.-Q.); denise.gastaldo@utoronto.ca (D.G.); 4Health Sciences Department (Occupational Therapy), Faculty of Health Sciences, University of A Coruña, 15006 A Coruña, Spain; 5Ciber Fragilidad y Envejecimiento Saludable (CIBERFES), 28029 Madrid, Spain; andreu.bover@uib.es; 6Department of Nursing and Physiotherapy, University of Illes Balears, 07122 Mallorca, Spain; 7Department of Philosophy and Social Work, University of Illes Balears, 07122 Mallorca, Spain; macarbonero@uib.es; 8Lawrence S. Bloomberg Faculty of Nursing, University of Toronto, Toronto, ON M5T 1P8, Canada

**Keywords:** migration, working women, workplace, social discrimination, racism, sexism, qualitative research, economic recession, Spain, caregiving

## Abstract

In Spain, most jobs available for Latin American immigrant women are in intimate labour (caregiving and domestic work). This work is usually performed under informal employment conditions. The objective of this study was to explain how the colonial logic mediates the experiences of Latin American women working in intimate labour in Spain, and the effects of such occupation on their health and wellbeing, using a decolonial theoretical framework. A multi-site secondary data analysis of qualitative data from four previous studies was performed utilizing 101 interviews with Latin American immigrant women working as caregivers in Spain. Three interwoven categories show how the dominant colonial logic in Spain creates low social status and precarious jobs, and naturalizes intimate labour as their métier while producing detrimental physical and psychosocial health consequences for these immigrant caregivers. The caregivers displayed several strategies to resist and navigate intimate labour and manage its negative impact on health. Respect and integration into the family for whom they work had a buffering effect, mediating the effects of working conditions on health and wellbeing. Based on our analysis, we suggest that employment, social, and health protection laws and strategies are needed to promote a positive working environment, and to reduce the impact of caregiving work for Latin American caregivers.

## 1. Introduction

Immigrants leave their countries of origin searching for a better quality of life for themselves and their families. The intensification of international travel, especially workers’ mobility, is directly related to the current process of economic globalization, which has been increasing the disparities between rich and poor nations [1]. For instance, the in immigrants residing in Spain has been remarkable. Since the mid-1980s, Spain has shifted from being a place of workers’ emigration to being an immigrant host country [2,3]. Foreign born people represented 1% of the population at the onset of the century (924,000 people in 2000), reaching its ceiling point 11 years later, when they represented 12% of the population (almost 6 million people). Those from Latin America configured the largest immigration group, being 42% of the total foreign-born population between 2001 and 2006 [4]: this phenomenon was coined by social scientists as the “Latinoamericanization” of immigration flows [5].

In Spain, the jobs available for immigrants are mainly in agriculture, travel industry, construction sector, and domestic work sector [6,7]. These jobs are seen as unattractive for Spanish citizens since most are low paid, precarious, and/or part of the informal economy. Being largely performed by immigrants for the last 20 years [6]. This trend is reflected among Latin American immigrants, for whom 37.6% of employment available for men is in the construction sector and 31.4% of employment available for women is in the domestic work sector [8]. Gender plays a key role in this occupational segregation, not only in Spain but globally. As Raijman, Schammah-Gesser, and Kemp [9] describe, the root cause for the increase in the number of immigrant women in such jobs is the industrialized nations’ growing need for caregivers and domestic workers. In Spain, caring for the elderly and/or people with disabilities has been identified as a key contemporary challenge for Spanish society [10]. Moreover, this need might be accentuated in contemporary Spain, where the welfare state is defined as an “implicit familism model” like other Southern European countries [11]. One of the key features of this model is insufficient services provided by the state for families in a context of increasing caring needs which assumes families (mainly native women who are part-time workers or not in the labour market), or a migrant woman [7,12] will “replace” female relatives [10], usually with minimal to no participation of men.

In the current global economic system, full-time employed women in high-income countries need a substitute for their domestic chores and caregiving activities, usually an immigrant woman [13]. In Spain, the roles of caregiving and house cleaning have been mainly delegated to Latin American women, as they come from former Spanish colonies and share cultural and language backgrounds [14].

As in all Europe, for caregivers a significant portion of job opportunities is in the informal market, offering precarious employment with no or minimal labour rights and social security [7,15]. Paradoxically, Spanish women’s liberation from caregiving and cleaning duties has been achieved in part at the expense of migrant women’s income, benefits, and health, most of them their former colonial subjects [10,16]. In particular, such exploitative relations are experienced by live-in caregivers working in private homes, who ensure the delivery of essential labour for child and elderly care [17].

### 1.1. A Decolonial Lens to Understand Latin American Women’s Domestic Work

Spain is a postcolonial space where contemporary forms of neocolonial ruling evolved from the imposition and disruption of colonial relations with Latin America within centuries of European colonialism [18]. Despite the elimination of colonial administration, colonial dynamics still operate in everyday relations [19]. The colonial logic of servilism is supported by racism and sexism [20,21]. Altogether, they are structural forces that cause injustice, dehumanisation, suffering, and health inequities [18].

This study is guided by decolonial theory [21,22], which aims at exposing the current effects of colonial logic with the ultimate aim of transforming them. For instance, in today’s imaginary about nations, discursive practices produce “the other” (e.g., non-first-world citizens, mixed-race people) as primitive and inferior, being more emotional and less rational, thus caring, loving and, on occasions, sexier [23]. From this perspective, the process of “civilizing” was a process of “domesticating” nature and people to extract resources, discarding “others” as members of their civilizations, such as the Maya and Aztec [24].

Domestication has been a central instrument of domination over the colonized. We argue that Latin American women working in the domestic sphere are “colonial immigrants” because although “never directly colonised by the metropolitan country they migrate to, at the time of arrival are “racialized” in similar ways to the ‘colonial/racial subjects of empire’” [19]. According to these colonial and capitalist logic, nature was understood as “female” and women were associated with nature. In this context, nature’s products have “no value” or are free and can be exploited, since the products of domestic labour is seen as a natural skill of women which remain mainly unpaid or poorly paid and undervalued [25].

Intimate labour (IL) is defined in this paper as all paid and unpaid domestic work, such as cleaning, caring for the elderly or children, and working as caretakers or homemakers. Traditionally, domestic work for women frequently included the role of live-in nannies and wet nurses, so intimacy has been historically taken for granted [26]. As a consequence, domestic spaces have been constituted by an intersection of stereotypical gender roles, naturalized servitude, under-recognized labour, low pay, ample amounts of free emotional and physical labour, and racialized and sexualized imageries of the “other”.

Caregiving delivers essential processes and products for human life and should be valued, but it also structures labour conditions and interpersonal relations because it has to be delivered in person all the time, commonly being exploited [27]. Such logic prevails in Latin America today, where, for example, in Brazil, the majority of domestic workers are mixed-race or black women and, in Mexico, they are indigenous women. The intimate ties created through caregiving activities reinforce hierarchy and asymmetry in micro social spaces [28], but domination is not only located in the space of the household, but also in the hierarchies of gender, race and class are experienced in society broadly.

### 1.2. Migration and Health

Migrants often experience health inequities as a consequence of changes in multiple social determinants of health and decrease in social position [29,30,31]. For instance, at the host country, immigrants might face discrimination, exploitation, difficult working and living conditions, high psychosocial stressors and social exclusion which negatively impact their health [29,32]. The specific challenges migrants encounter led scholars and international organizations to claim that migration is in itself a social determinant of health [33,34].

According to the World Health Organization (WHO) Social Determinants of Health (SDoH) Framework, social position is core to people’s health; the search for better socioeconomic status (SES) motivates most people’s migratory journeys and their acceptance of precarious working conditions [33,34,35]. Internationally, migrant workers are disproportionately affected by, and exposed to, health risks at work and this is obvious for undocumented migrants who are typically employed in “3-D” jobs (dangerous, dirty and degrading) in under-regulated sectors, such as domestic and agricultural work [32,36,37]. In Spain, researchers have documented immigrants’ poor quality of life as well as physical and mental suffering as consequences of their precarious work [15,38].

For immigrants who arrived in Spain before 2006, during the period between 2006 and 2012, their health status has worsened compared to natives [39]. The loss of the healthy immigrant effect, according to which migrants may have a better health status than the natives—because those in the best health are who migrate—may also be related to the impact of the 2008 economic crisis [40]. Poorer health outcomes for immigrant women working in domestic jobs have been reported when compared to their native peers. Research shows that these women have the highest risk of unfavorable factors for health at work, such as countless work demands, little social support or low esteem [7,10,32,41]; they have greater levels of stress [42]; worst self-perceived health [43]; report higher levels of self-perceived discrimination [44] and are the group with the highest prevalence of mental health problems [45]. Consistently, immigrant female workers score very low in the SF-36 questionnaire dimensions of physical and emotional roles (used to measure the perceived quality of life) [38], and their narratives illustrate the concurrent negative impact on their physical (e.g., muscle aches), mental (e.g., desperation) and social (e.g., isolation) wellbeing [46]. Additionally, their low SES as immigrant women working in IL has been associated with difficulties in accessing and using social and health care services in Spain [47].

Despite the identification of these health inequities for this group, little is known about the mechanisms by which the SDoH (specifically, the SES of IL workers) impact these women’s health and wellbeing. Therefore, in this study, we address the health and wellbeing of Latin American (race/ethnicity/nationality) immigrant women (gender) working as domestic labour (occupation/class) in Spain, using a decolonial framework [19] in the context of the migration flow that generated the so-called “Latinoamericanization” of immigration to Spain in the first decades of the 2000s [24,48,49]. In summary, the objective of this study is to explain how the colonial logic mediates the experiences of Latin American women working in IL in Spain and the effects of such occupation for their health and wellbeing.

## 2. Methodology

This is a multisite qualitative study based on secondary data analysis from four critical qualitative health studies [46,47,50,51] aimed to better understand diverse aspects of the health and wellbeing of Latin American immigrant domestic workers living in six Spanish regions (Table 1).

Multisite qualitative studies allow to revisit and analyze the raw data of a set of studies to answer a new research question and are suited to increase transferability and trustworthiness of contextually relevant findings [52]. We used Jenkins [52] multisite qualitative analysis to generate new understandings of immigrants’ circumstances in each site, transcending the specificities of each location while keeping the relevant contextual information allowing to identify the health consequences of performing IL and working in a country operating under a neocolonial logic for Latin American immigrant women. The original studies could be re-analysed due to their shared critical onto-epistemological position that guided their theoretical framework and methodologies, and the substantive interest in participants’ health and wellbeing (Table 1).

In total, 101 individual semi-structured interviews performed between 2009 and 2015 were included.

### 2.1. Participants

Intimate labour for many participants entailed cycles of live-in and live-out workers. Combined, participants were Latin American domestic workers from 11 countries (Colombia, Argentina, Ecuador, Chile, Bolivia, Brazil, Uruguay, Dominican Republic, Peru, Cuba, Venezuela). Participants’ length of stay in Spain ranged from two years to 14 years, with precarious and legal migratory status, an age range from 20 to 52, living both in rural and urban settings (Table 1).

### 2.2. Recruitment

Participants were recruited in six Spanish regions where the studies were conducted, first contacting Latin American immigrant associations and cultural mediators to recruit, and second using a snowballing technique, asking participants to provide contacts for other women who would fit the inclusion criteria.

### 2.3. Data Generation

Interview guides were prepared after a review of the literature and according to the objectives of each study (Table 1). Semi-structured interviews were conducted in participants’ homes or other spaces selected by participants (e.g., public parks or shelters), during 45 to 90 min by the principal investigators of each project (A.B.-B., M.A.C., M.G.-S., N.R.-Q.), by co-investigators, or research assistants. All the interviews were conducted in Spanish as the mother tongue of both interviewees and interviewers, except Brazilian women, for whom fluency in Spanish was required as an inclusion criterion.

Interviews were digitally recorded and transcribed verbatim. During interviews, the interviewers made fieldwork notes which were also included in the data analysis.

### 2.4. Data Analysis

We conducted the data analysis using Jenkins et al.’s [52] three-phase approach to multisite qualitative analysis. First, each original study was analysed separately by its principal investigator (within-site analysis). The original four studies used open coding as their first step in their respective analysis. Each PI used such coding and the original transcripts in an iterative process of back and forth, to refine such coding according to the new research question. Secondly, two authors (E.B.-V. and D.G.) conducted the *between-site analysis*, searching for variations within the data originated in the four sets of data. They did so, after two authors (A.B.-B. and M.A.C.), re-analysed their data and created a preliminary list of codes. This coding scheme and the theoretical framework aforementioned (i.e., elements related to power relations, colonial logics and its effects in health and wellbeing, among others) were then used to analyse all sets of data. Next, following an inductive process, when necessary, preliminary codes were modified to refine the analysis, creating new open/emerging codes. Subsequently, all the codes were grouped under subcategories and categories. Finally, the PI of the four studies contributed to the second within-site analysis (third phase) by returning to the original studies’ data and ensuring the overarching categories generated in phase two resonated with the data and all relevant information was included. When discrepancies emerged, authors in charge of phase two (E.B.-V., D.G.) and the principal investigators of the original studies clarified points, reaching agreement.

### 2.5. Quality Criteria

Criteria for quality data analysis was used to guarantee the rigour of the study. We adapted the three-phase approach for multisite qualitative studies and had an objective for the analysis and a theoretical framework to guide the process [53]. We also benefited from a large team of analysts working in different phases. Researchers with different backgrounds (nursing, occupational therapy/science, social work, sociology, public health, and gender studies) with shared theoretical grounding participated in different parts of the analytical process. The identification of quotes from all studies helped to clarify how the authors’ interpretations were grounded in data and supported shared understanding. Spanish was used for data analysis; translation into English only took place when the results were drafted. The English translation of the quotes was double-checked by a native translator fluent in both languages.

### 2.6. Ethics Approval

Participation in each original study was voluntary, and a consent form authorizing the use of the data with research purposes was signed by all participants. The appropriate ethics committee approved each project: CE06/2013 (Comision de Bioética de Illes Balears: FOU194, Comitè Ètic d’Investigació Clínica de Lleida: CEIC, Ethical Comitte of the University of A Coruña: CE-UDC).

## 3. Results

Three interwoven categories show how gender, ethnicity, and working conditions, linked to servilism and discriminatory practices, impacted participants’ health. Category 1 explains IL as a work niche for immigrant Latin American women in Spain, where immigrant women’s caregiving is naturalized. Category 2 illustrates participants’ working conditions were mainly underpinned by neocolonial relationships and their effects on health. Personal relationships mediated the effect of working conditions on health and wellbeing: if relationships were based on exploitation, servilism, and discrimination, the health effects would be worse. Nevertheless, the experience of respect and integration in the family had a buffering effect. Category 3 describes women’s strategies to resist and navigate IL and manage its negative impact on health.

### 3.1. Intimate Labour as the Gendered and Racialized Métier of Immigrant Latin American Women in Spain

When participants arrived in Spain, they found the main job available to them was as caregivers and/or cleaners, regardless of their previous work experience or abilities. For instance, Lucía (Study 4) explains how “[she] *had never experienced what it looked like seeking a job here* [Spain]. *And, of course, the only* [thing] *they* [employers, employment support offices, etc.], [offered as an option] *was “domestic work*” (Study 4, Lucia).

Participants’ stories conveyed widespread stereotypical ideas about Latin American women: they are better equipped for IL because they respect and value family and they are perceived by employers as more loving and submissive than Spanish women. Participants reported specific attributes that make an immigrant woman desirable for IL: being obedient, polite, respectful, helpful, caring, docile, and submissive, as well willing and able to work around-the-clock.

At times, even the workers themselves, internalized this belief and displayed these skills as an asset to get or retain a job in this sector. For example, they explained that, due to their culture, [they] *are more caring than Spanish women*, which would benefit their performance as IL workers. This naturalized gendered and racialized conception of IL and Latin American women’ skills was presented by some participants as an advantage compared to their male counterparts: “*Women find jobs* [in Spain] *more easily than men* (…) *because there are more* [jobs] *to take care of grannies, to take care of children, for cleaning, right? There are so many options for women, while for men there is* [in Spain] *just one job* [construction work]” (Study 1, Yolanda).

Nevertheless, these job opportunities, limited to diverse forms of IL, clashed with their previous work experiences. Most of the participants had worked in jobs that needed higher qualifications back in their countries and did not expect to become IL workers: “*Picture yourself cleaning bathrooms, mopping floors. I have worked in a construction company in Argentina, in charge of the finances. Then [as an IL worker], of course, I felt badly, it was not ok*” (Study 2, Amanda). Participants were aware of the mismatch between their professional abilities and the aforementioned naturalised IL requirements, which astonished them once they first realised “*we come here [Spain] and had to clean or take care of the elderly or children” “(…) we were not ready for this, you know? Because one thinks ‘well, I have certain skills for other things’*” (Study 1, Zuleima).

Participants inform IL is the only option for them in Spain. Sometimes it is used as a temporary strategy to move into a better job, regularize their migratory status or simply as a survival job: “*I put up with a lot to get the papers (Study 1, Eliana)*.” However, they also explained they do not always achieve that goal: “*I do not need a person with papers [regular migratory status] here—she [the potential employer] told me— “I am not here to make contracts (...) a person who comes to take care of my mum does not need papers*” (Study 2, Teófila).

### 3.2. IL Working Conditions Based on Colonial Relationships: Consequences for Latin American Women’s Health and Wellbeing

#### 3.2.1. IL as Servilism: Trapped in a Context of Vulnerability, Exploitation and Abuse

Specific attributes that make an immigrant woman a desirable domestic worker not only contribute to Latin American immigrant workers joining IL jobs, but also impact on the interpersonal dynamics in the workplace. Once participants accessed these IL occupations, they mainly reported precarious, abusive, exploitative conditions or under recognition. For instance, they described situations of working extra hours and performing additional tasks with no increase in salary or any other form of compensation or recognition:
“*When they employ you, they employ you as a babysitter…but that is a lie. Of course, you have to babysit, but you have to manage and run the house chores, iron, vacuum, take the kid to her school, shower her, feed her, cook for them* [family] *for lunch time when they come to eat (…) that’s it, they exploit you and it is an atrocity what they pay: nothing*”(Study 1, Lis)

Excessive demands and/or supervision by employers is perceived as a demonstration of employers’ superior status, which in turn makes Latin American women feel enslaved and subjected to servile relations: “*she* [employer] *would make me clean the floors on my knees, with a brush, you know? And I thought: “the mop exists already* [so], *why?*” (Study 1, Rocio)

“*Actually, they treat you like a slave (...) that is what bothered me, being told that ‘you have to be at my entire disposal’, it is like saying ‘you have to be here enslaved, doing what I say’. And, of course, because you have no other job, you have nothing else, well, it is kind of accepting it, of course, you have no option but saying yes*”(Study 4, Lucía)

These situations cannot be reported to authorities, since IL usually occurs in the context of the informal economy, without a legal recognition of employment or workers’ rights. These power abuses are naturalised by employers as part of IL, and seen by most women as part of the abusive employer–employee relationship for the specific context of being an immigrant woman in IL in Spain: “*Employers think that they can command people with their money “*(Study 1, Teófila).

Participants explain how employers, aware of workers’ precarious immigration status, explicitly or implicitly threaten them (Study 3). This participant described her employer’s words to her: “*You need the money. Then I pay you. You have to send* [back home] *it* [money] *for your kids. You shut up (...) you need the money, then you work. If you leave here* [this job], *where are you going to get the money to send to your country?*” (Study 3, Rosalba)

Participants “*have to endure and endure*” (Study 4, Maria) the negative impact of servilism in their lives, since the alternative is “*nothing*”: “*women working as a domestic worker, if they fire you, you have no right to unemployment benefits(...) they fire you and you are left with nothing*” (Study 4, Lucía).

All these make participants feel they are not treated as human beings “*they* [employers] *are using you*” (Study 1, Zuleima). For instance, “*They [employers] focus only on themselves. You have come from your country, but they never ask you if you are all right, how you are or if you found what you were looking for*” (Study 2, Cecilia).

The dehumanisation described seems to be exacerbated when the IL is performed as live-in workers, caring for elderly or disabled people. These live-in workers share house with their employers and/or care-recipients, and must be available 24 h a day to meet employers’ demands: “*the husband* [in the family] *had a little bell and every time I heard it I thought he wanted to urinate and I would get up, sometimes four times in the night, it was killing me*” (Study 1, Dinora).

Most of the participants have worked as live-in workers. Consistently, they explained that these jobs are the hardest and would try to search for another option of IL whenever possible:
“*because, of course, when you are a live-in worker, people abuse. The bosses, you know? And then they want you to do more hours than those, those that you are supposed to work. And one friend worked till midnight. I wouldn’t have endured it, not even the first day*”.(Study 4, Elena)

Since immigrant women in the IL are not perceived as people with human rights, basic rights such as privacy and personal space are disregarded. Some participants had to “[sleep] *in the same bedroom* [of the care recipient] *(...) they* [employers] *say I have to sleep there. (...) If she* [care-recipient] *sleeps, I do not sleep; if I sleep, she doesn’t fall asleep*” (Study 1, Isabel).

Dehumanisation is combined with the naturalisation of all forms of caregiving tasks as female work. Expected IL tasks for Latin American women are: cleaning, cooking, emotional support/showing affection, and personal and intimate hygiene, in addition to specific health care tasks, such as healing wounds, giving injections or handling devices to mobilize disabled people.

“*I had to take care of an old person and I had to do things that I had never done in my country, such as changing a diaper. And he was ill, the prostate. I was not expecting it*”.(Study 1, Vanesa)

Despite the specialised nature of many of these tasks, with no connection to their previous work experiences, some participants were frustrated because, as workers, they did not receive any specific training for those activities or appropriate supplies, creating hazardous situations:
“*I asked them* [employers] *for gloves, and they didn’t want to provide them. But I always had to do tooth brushing, and all that. I didn’t like it because the person had many wounds (...) and I had to use the hoist to move [the person] in bed.*”.(Study 1, Nuria)

There was a marked contradiction between employers’ naturalisation of all care activities as proper work for the caregivers and the workers’ understanding of IL as a job which should entail respect for workers’ rights.

The dynamics and conditions of IL as servilism, mediated everyday relationships between employers and participants, generating what participants call “abuse”. This abuse took different forms that sometimes overlap; for example, the aforementioned working conditions, employers’ threats, disregard for basic needs, such as private space, sleeping hours or safety, among others.

“*the abuse had already started, [the employer] already started to abuse, to treat me badly. I went to sleep late, I had to get up early because I woke up the kids, got them dressed, I virtually did everything*”.(Study 1, Eliana)

Often, forms of abuse included explicit forms of violence, such as physical violence. For instance, the children one participant cared for would hit her with no repercussion or limits set by the parent: *The kids hit you… and it was very, very… [frustrating]. (...) You come here, you are the employee and that’s it (...). Children mistreated me and their mother…* [consented] (Study 3, Rosalba).

Participants also reported attempts of sexual abuse from male employers. Some of them expected sexual intercourse as part of the job. Participants clearly identified that being an immigrant woman made them more vulnerable to suffer such abuses. They explained Spanish men sexualized Latin American women due to their exotic condition.

“*(...) due to being an immigrant, due to being a woman. They think every Brazilian woman comes here to be a prostitute, you know? One day, I went for cleaning [a house], and a guy told me: “if you come and spend time with me [have sex], I will give you 50 [euros]”. Then I said: “No. I do not do that” and he said: “but everyone* [immigrant woman] *does it*”.(Study 3, Lilian)

These women felt unprotected towards abuses, especially if they did not have a work visa in Spain:
*My boss’ son wanted me to... he wanted to take advantage of me* [sexually] *(...) But I could not do anything, I have no papers. Who can I go to? I was a bit afraid. Who can I go to ask for help? I … I didn’t know what to do. I was also beginning* [in this job], *I didn’t know what to do, what to say, out of embarrassment*.(Study 1, Claudia)

All these forms of abuse are underpinned by the intersection of different and intertwined oppression categories: race/ethnicity, gender, and social class. Some participants clearly identified discrimination based on race/ethnicity, gender, and social class as the root cause of their experiences of abuse, while others referred to them as the generic term “discrimination” or “racism”: “*there was some racism (...) they* [employers] *humiliated me, mostly*” (Study 3, Estela).

#### 3.2.2. Neocolonial Logic: Health Consequences for Latin American Women Working in IL

IL is a form of servilism that negatively impacts Latin American women’s health and wellbeing. Participants described a myriad of negative physical, psychological and social health consequences. Table 2 presents the main health consequences described by the participants.

Latin American women attributed many symptoms they experienced to the tasks they had to carry out; for instance, the lack of specific training or injury prevention, repetitive physical strain that cleaning and mobilising disabled people required, and other very demanding physical activities:
“*they are elderly people that are sick, in bed. Then, you are going to be exhausted because you have to lift them, dress them, lift them again, transfer them to the chair, move her from the chair to another place…*”.(Study 1, Alba)

These physical demands are paired with lack of time for personal rest and physical recovery, as one participant explains: “*I did not rest, not even…, not even half an hour, I didn’t rest because I had too much work*”, and difficulties for eating properly: “*I had no time to eat. I started to slim down and I got sick. I got sick and I was hospitalized for a month (...) you cannot live without sleeping … I was debilitated, very skinny*” (Study 1, Isabel).

Despite the importance of IL physical consequences, for participants the worst impact was on their mental and emotional health. Sometimes, the physical consequences are understood as part of the job: “*Well, it is normal, when you take care of children, you will have back pain, some ache in your legs. It does affect you (...)*” (Study 1, Zuleima). This statement shows how participants tend to normalize the physical impact of IL. Nevertheless, they went on to emphasize the emotional burden as the most significant consequence: “*(...) It does affect you, but I think what affects you the most is the emotional [aspect]*” (Study 1, Zuleima)*. “For your mind, it is much worse*” (Study 3, Rosalba).

Table 2 summarizes the mental health issues experienced by participants. IL demands on mental health were heavy and difficult to deal with. Some examples were living and taking care of people with cognitive impairment, the death of clients or managing all the abusive conditions. Participants expressed feeling burnout, stressed and depressed as consequences on their mental health: “*She was a person [employer] one of those always quarreling. At the end of the day, even though you did nothing, you end up psychologically ...pufff! shattered*” (Study 1, Alba).

The emotional involvement of caring as part of IL, that is the personal and intimate closeness of workers with the persons they cared for, is highlighted as a key stressor often neglected by employers, and with a high impact on participants’ emotional wellbeing:
“*because they* [employers] *pay you to clean the glass, and they pay you to scrub the pots. But becoming fond of the person you care for, the psychological support that you give [to that person], nobody pays that. That makes you leave [the job] with your head like a drum [exhausted]*”.(Study 4, Olivia)

The isolation and lack of freedom for those live-in workers also contribute to these negative feelings and to a lack of social participation.

“*in the beginning I had a rough time, because from having freedom to go outside whenever I wanted, now I couldn’t do it. I had to be with the lady from Monday to Saturday. Saturday at lunchtime my day off started till Sunday evening. But from Monday to Saturday, a whole week inside the house. Well, I could go out to buy bread or something she needed… the change was super shocking, it was really difficult (...). The first two weeks I cried everyday...because it was heavy*”.(Study 1, Carolina)

#### 3.2.3. Perceived Positive Elements Buffering IL Health Consequences

In some cases, some participants, in some of the jobs they have had, encountered better working conditions or perceived some positive elements IL could bring to their lives. For example, good relationships with employers or fair remuneration, among others, were positive elements for participants. Since Latin American women were aware of the vulnerability that the combination of IL and being an immigrant woman entailed, when experiencing these positive aspects, they felt lucky or thankful. These positive features buffered the negative impact of being an immigrant woman performing IL but do not change the intrinsic colonial dynamics that characterised IL for participants.

Most of the positive aspects of IL pivoted around having good relationships with employers and their care recipients (seniors or children): “*Many times we are fine, happy. There are also very good people, I have had [as employers] very good people and they know me and love me very much. I am very grateful for those people. But... there is a bit of everything*.” (Study 4, Sara)

These positive relationships were experienced as different forms of support. For instance, “*they* [employers] *arranged my paperwork* [migratory status], *they helped me a lot.*” (Study 1: Silvia). Some employers also fostered the opportunity to negotiate working conditions: “*the lady* [employer] *I am with now; she gave me the freedom to cook what I want. I call her by her name; I do not call her ‘Mrs’. She told me to sit on the couch with her, not like other people who* [want to keep] *distance*” (Study 1, Dinora).

Another IL feature highlighted as positive by participants was the emotional ties with the employers’ families. Workers experienced affective interdependence with their employers and their relatives. A participant told that her employer described her job in these terms:
“*I do not pay for your time; I pay for the love that you give to my children and for the way you are with my children ... that is what I pay you for.*”(Study 3, Piedad)
“*[before, in the country of origin] I had a good job, totally impersonal. And here I learned to be a person and to have feelings for people who I didn’t know, to love people, people who, without knowing who you are, they do everything for you. Simply, because the girl [employer] thought that I was a good woman. The ties that you create are of affection*.”(Study 2, Mariana)

The care work provided by participants becomes a labour of love when bonds of affection were established with the people cared for (children or seniors); a good relationship with their employers makes them feel like family.

“*the children always cried with the possibility of me returning to Colombia... they got sad, they asked why would I leave someday. Then, it is like that really impacted me, but I got adapted [to the job in Spain] easily because of them. Because I never felt neglected, I always was one member more of the family.*”(Study 1, Piedad)

Relationships with the employing family based on love were especially positive for the women who migrated alone: “*And, if I need support, or if I need to talk, I talk to them and for me that is... If I do not want to feel lonely, because that’s what one feels the most here [employers are there] (she cries). Especially, because I have no family here.*” (Study 1, Silvia)

Being involved in a loving family in the host country helped Latin American women to adapt to the new country and prevented them to feel alone missing their families abroad—*” she always saw me as a granddaughter. It was like finding a family*” (Study 2, Mariana)—or even to compensate at some extent the lack of their family: “*I took care of them as if they were my kids.*” (Study 1, Eliana)

Participants described situations when mutual affection between them and the people they cared for grew on them, compensating some of the IL sacrifices. IL was then perceived not as a regular job with a predefined schedule, but something more meaningful, not only fueled by remuneration, but by their sense of responsibility and compassion for the care recipients. This might entail putting forward their employers’ requests and postponing their own needs.

“*There are other times that it very much makes up to you because if you are a good carer (...) the children will love you very much and that helps you very much. For example, when I had my lows (..) the love they gave me. One smile made it all up.*”(Study 1, Margarita)

Despite the positive aspects of having good relationships with the employer’s family, IL would still entail isolation and personal considerable emotional labour to support care recipients, what can be detrimental to participants’ emotional wellbeing. For instance, dealing with the specificities of cognitive deterioration such as mood alterations: “*I work with an old lady, right? I am fine with her, wonderfully. Although her temperament impacts me, right? That change in her temperament it impacts on me.” (Study 1, Lisethe) or* witnessing their deterioration or even death: “*that was the first sadness here, it was when the lady died” (Study 1, Silvia)*. “*It made me suffer, you know? (...) it was very hard to see her everyday there, she lasted twenty-one days without eating and then she died.*” (Study 1, Mercedes)

### 3.3. Latin American Women Forms of Resistance and Negotiating IL Struggles

The interviewees described multiple forms of oppression, but also resistance. Eight situations were identified in Latin American women’s accounts, corresponding to the diverse forms of resistance and negotiation of their situations (Table 3). These forms are not fixed nor exclusive. One person might utilize all of them or just a few, and they can even overlap in time.

Participants endured work for economic reasons, especially if they had children, despite being aware of the negative consequences of IL for their health. Sometimes, they felt forced to be silent or avoid expressing their psychological or physical suffering when: (1) the job entailed abusive conditions or exploitative situations, they refused harmful jobs to protect their dignity, personal value, and made strong physical efforts to protect their health; (2) sometimes, their vulnerable situation in Spain forced them to accept IL, that could include several forms of abuse. Nevertheless, within this scenario, participants were adamant to avoid prostitution; (3) they negotiated with employers the hours and days off in their jobs as live-in domestic workers so as not to feel cloistered and try to compensate for the social isolation of interim work; (4) they thought about quitting jobs that were harmful to them, especially part-time jobs; (5) they quitted jobs exploiting them to protect their dignity, personal value and health, taking into account that some of them had several jobs, which increased exhaustion and health risks; (6) they considered going back to college/school to be able to access other jobs and improve their situation, aware of the triple work shift studies would represent; (7) participants were aware of the limited job opportunities outside IL and even compared themselves with others living worse situations as a strategy to navigate the struggles they face; and (8) finally, they contemplated returning to their countries, as a result of the physical and mental distress generated by their work.

## 4. Discussion

The main argument of this paper is that currently the dominant neocolonial logic in Spain creates low social status and precarious work for Latin American women, it naturalizes intimate labour as their métier, while producing detrimental physical and psychosocial health consequences for these immigrant domestic workers. While working conditions and salaries have deteriorated in many sectors in Spain after the 2008 crisis, we believe the level of exploitation these women experience is unique due to the compounding effects of migration, racialization, gendered working conditions as social determinants of health.

In Spain, immigrant women’s triple discrimination has been described by several authors [7,54,55]. The process of naturalization of IL as fitting for Latin American women, who are discriminated against for being immigrants and caregivers performing work that is socially invisibilized and not valued, captures most of the participants’ experiences. Their competence as carers is naturalized, their previous qualifications are ignored or not valued, and the skills they bring to the country are invisibilized. Their status as immigrants and their economic need make them quite vulnerable to exploitation, precarious working conditions, and low pay [15]. Additionally, being overqualified for domestic work made participants feel frustrated for not being able to find jobs with better conditions and matching their expertise. Previous studies in Spain considered that women’s employment expectations are not being fulfilled [38]. Despite participants’ willingness to get better jobs, similarly to what has been reported in recent studies, we found out that many of them took domestic work as a temporarily first step into the Spanish labour market, as a gateway to regularization of their migratory status or their only income opportunity when the post-2008 crisis cramped job availability [46]. Nevertheless, because care work usually occurs under informal employment conditions, these jobs traditionally do not permit immigrant women to obtain legal status [16,56]. In fact, in Spain, female migrant manual workers are more likely to work without a contract than migrant men or non-migrant women [57,58].

Domestic work is not only devalued because it is considered as non-productive labour (even in its waged form), but also because those doing the work are stigmatized for their foreignness [59,60]. Being women and “Latinas” legitimized labour segmentation and naturalized domestic activities as their occupation [61,62]. Caring attributes, that are perceived to be innate to women, are part of employers’ social imaginary and portrays immigrant women as pleasant and submissive, therefore capable of the emotionally demanding labour of care [59,63]. The reproductive work is often not recognized as employment because (1) it takes place in a private home and (2) the tasks that domestic workers do—cleaning, cooking and caring for seniors and children—are associated with women’s “natural” expressions of love for their families [56,64]. Other authors explain that management of feelings is generally not valued as labour, despite its relevance in sustaining social relations because of being normally privatized and feminized [65]. The practices of IL within home space reflect and reinforce larger structural inequities regarding ‘race’, gender, culture and citizenship in both national and transnational contexts [66]. Migrant women who are separated from one’s own culture could experience a sense of guilt and fracture, which combined with precarious working conditions, seems to be associated with intense emotional distress, depression, and psychopathological symptoms [67,68].

Our analysis reveals that naturalising IL as an occupation for immigrant women leads to precarious working conditions and poor health. Our finding corroborates other studies that have found that migrants are more likely to face adverse working conditions and employment arrangements that possibly will place them at increased risk for health problems [41]. Moreover, previous studies with a migrant population in Spain indicated a higher prevalence of occupational accidents, especially among women, due to greater exposure and less prevention of occupational risks in less qualified jobs and when extended working hours occur [61,69].

Latin American women in IL embodied the consequences of using their bodies as their working instrument [70]. Physical effort and constant dedication cleaning and caring result in women experiencing pain and fatigue. When employers disregard Latin American women workers’ healthcare needs—for example, by making it difficult for them to access the healthcare system due to incompatibility of schedules with their jobs, especially for women who are live-in caregivers—they are violating their rights as workers, relegating them to a secondary position of importance vis-à-vis the people they care for. In fact, Spain faces a global care paradox: the women working for those in need of care (seniors, children, people with disabilities, etc.) who are traditionally situated in wealthier areas of the world, are not cared for, and their families—whether in the host or home country—miss fundamental care resources [13,41,68,71].

The perception of a low position in the social gradient can produce high levels of stress or unhealthy behaviours that are detrimental to health [72]. As Gutiérrez Rodríguez [49] said, their bodies must carry ‘the disgust’ and ‘dehumanizing effects of racism’ expressed in the devaluation of intimate labour. Migrant workers are more exposed than non-migrants to adverse psychosocial conditions and risks, so they are more at risk of suffering mental disorders [57,73]. Several mental health issues (e.g., mood instability or depression) related to work conditions (for example, exploitation, low levels of control) are common among female immigrant workers [49,57]. We have identified elements that explain this phenomenon; care workers often are isolated in private homes (which is a unique health circumstance). They suffer due to the oppressive neocolonial logics of exploitation and abuse that shape interactions in such private space [41]. Participants in this study reported often having experienced discrimination in forms of abuse, such as delayed payment, withholding food or sexual harassment, coinciding with a similar study in Portugal [74]. In addition, Latin American housekeepers in hotels are also vulnerable to abuses including disrespect, unfair job demands, and verbal abuse by managers [75]. Nevertheless, live-in care workers who perform labour in private households, as the majority of participants have experienced, are especially vulnerable to the demands of their employers, which exposed them to a higher risk of economic, social, physical and sexual abuse [32,41,60,76]. As Latin American women participating in this study reported, host country employers’ proposals linked to prostitution are explained because they thought the body of women in the form of sexual services could be included in monetary exchange. Similar to our findings, other authors such as Ong [77] explained cases of seduction or rape by male employers among Asian domestic workers. Such patriarchal and neocolonial imaginary reflects structural violence towards foreigner and racialized women [46]. The Latin American stereotype is the porno-tropical colonial idea of an over-sexualized woman, more sexual and passionate than Europeans and readily available for sex [23]. Racialized women’s bodies seem to be perceived as always available as an endless source of domestic and sexual labour [49], an inextinguishable source of free or poorly paid caregiving and comfort for Europeans. Making all these effects visible also challenges global neoliberal dynamics [68].

Latin American immigrant women’s forms of resistance regarding IL are struggles against exploitation [59]. The significance of family connections and networks for immigrant caregivers emerged as a mechanism to cope with the demands of caregiving and to relieve the burden of care [78]. Participants themselves incorporated discourses of domesticity, believing that employers are like their family through an affectionate language. According to Lynch [79], the intimate labour Latin American immigrant women perform fits into general care work, but also overlaps with the labour of love that maintains care relations. According to Casado-Mejía [80], positive and egalitarian interpersonal relationships with employers are a protective factor for live-in female immigrant caregivers’ health. However, some authors understood such cordial behaviour can function as another way to intensify domination, especially regarding the inequities experienced by women at IL [81]. For example, the expression used to name the worker as a ‘member of the family’—often used by both the employer and the employee, could be seen as an ideological mechanism or strategy intrinsically linked to unequal power relations [28]. This situation could bring some benefits to both parties: the worker feels appreciated and the female employers partially neutralize the unpleasant feeling of having a stranger in their intimate métier, their homes [28]. Moreover, we argue that when Latin American women understood working for Spanish employers as a situation of mutual benefit (an interchange of “favours” or codependence), in which immigrant women replaced employers caring for their relatives and, at the same time, employers created a job position, they are internalizing and normalizing inferiority and a highly unequal social position [59]. This mutual dependency of women is called “politics of affect” and it keeps the fundamental aspects of life being delivered for men, wealthier women and their family members [49].

Resistance to return to their original countries may be explained because returning might be perceived as losing opportunities for future better quality of life regarding work stability and social guarantees, as a recent study with Colombian migrants who returned from Spain indicated [82]. Accepting to return is a frustrating decision for migrants, and the process of re-adaptation in their country of origin represents a rupture in their health, employment and work expectations [82].

### 4.1. Implications

This study shows that private spaces are constituted by international neocolonial relations, rather than being private, interpersonal spaces only. We believe that changes in policy and practice depend on overcoming capitalist, patriarchal, and colonial relations in Europe [83]. The health outcomes described here are the consequence of an entanglement of socio-economic-political relations that are characterized by exploitative relations between Spanish employers and Latin American immigrant women as caregivers [32]. In this context, there are at least three levels for implications of our study to be considered: changing mentalities and ideas, creating programs, and delivering care.

At the level of the collective imagination, the naturalization of Latin American women as caregivers who are emotionally available for caring all the time is at the basis of exploitative relationships. Implicit ideas of indigenous people as primitive, ‘race’ as a form of inferiority, and women as natural caregivers has created a platform for precarious working conditions, poor pay, discrimination, moral assault, among other conditions that make immigrant women experience illness and suffering. These neocolonial relations have to be problematized and the human rights of these caregivers have to be affirmed. Policies and programs to support paid caregivers should address these inequities, but we believe a social awareness movement is needed to de-naturalize IL as ‘the occupation’ for Latin American women in Spain.

At the level of policy and program development, it is important to acknowledge the effects of living in a position of social discrimination. It is known that high levels of subjective social status can provide people with psychological resources, such as increased self-esteem, security, hope, and feelings of control, which are important resources for health [84]. In the case of immigrant Latin American women working as caregivers, this means policies and programs should assure dignity and labour rights are achieved in all workplaces.

In addition, immigrant caregivers’ occupational health needs should be addressed in labour policies. Given the factors previously mentioned (i.e., emotional implication of IL), workers in IL are in great need of mental health services and other types of care [31]. The occupational health needs of immigrant workers must be addressed at the workplace and health care services levels while improving the enforcement of existing health and safety regulations [15]. Detecting specific health needs for Latin American immigrant women who work in IL at the primary health care level could help to guide interventions aimed at promoting and protecting the health of immigrant caregivers.

Finally, a better understanding of immigrant caregivers’ perceptions related to their health experiences is fundamental to design content and intervention strategies adapted to them in Spain [52]. For example, to allow the development of psychosocial programs to support working immigrant women as caregivers and new research to explore the specific situation of this vulnerable group in relation to the COVID-19 pandemic.

### 4.2. Limitations and Strengths

This multisite qualitative analysis has helped to generate contextualised knowledge that can be used for program/policy level discussions and actions. Although the results are not able to assure causality, they can be used to theorize about the health consequences of IL in Spain and similar settings from women’s perspective.

It should be taken into account that in this multisite qualitative study: (1) the original studies took place 6 years timeframe (2011–C15), (2) participants’ diverse origin countries and associated cultures may have nuanced some of their experiences, and (3) some variables such as educational level were not registered. Nevertheless, we consider that: (1) all interviews were conducted after the 2008 crisis began, in the same context of increasing vulnerability for immigrant workers; and (2) diversity in participants’ nationalities helped us to identify the shared experiences of Latin American women working in IL, which is a Spanish social construct that supports a shared social imaginary which is not directly related to any specific country. Similarly, the different geographic locations of the original studies in Spain allowed us to identify commonalities regardless of the diverse Spanish regional contexts, contributing to the transferability of our findings. Including 101 interviews strengthened data saturation about IL, despite the fact that it could not have been achieved in every substudy. Including studies that shared similar research aims, theoretical frameworks, methods, and participants’ profiles generated extensive, comparable and rich data, and contributed to a robust methodological design.

The inability to perform solid member checking in this design could be considered a limitation in reliability. Nevertheless, different measures were taken in this study to guarantee reliability of qualitative research, such as triangulation, a detailed context of the study, and use of literal citations. Regarding our positionality as researchers, all authors are from Spain, with the exception of one Latin American researcher (DG), whose insider’s perspectives allowed us to question our analysis and deepen our interpretations.

## 5. Conclusions

This paper explained the relations of domination in neocolonial societies like Spain. The findings revealed that intimacy is a space regulated by both explicit rules (e.g., government, laws, policies) and by specific social discourses (e.g., what is understood as intimate, as private, and as public). The delimitation of private spaces is a fundamental element to understand the impact of the migration process on physical, mental, and social health.

## Figures and Tables

**Table 1 ijerph-17-08278-t001:** Characteristics of the sample and original studies.

Characteristic	Study 1	Study 2	Study 3	Study 4
Geographical zone	Balearic Islands Catalonia Basque Country Madrid Canary Islands	Balearic Islands	Catalonia	Galicia
*n* of participants	51	24	12	14
Nationality	Colombia Bolivia Venezuela Argentina Guatemala	Colombia Bolivia Chile Argentina Brazil Dominican Republic Ecuador Uruguay Peru	Colombia Bolivia Chile Brazil Dominican Republic	Bolivia Venezuela Chile Cuba Dominican Republic
Status	Documented and undocumented	Documented and undocumented	Undocumented	Documented and undocumented
Age range	25–45	20–50	19–52	36–42
Work location	Urban	Urban	Urban and Rural	Urban and Rural
Year	2010–2012	2009–2011	2011	2013–2015
Objectives	To analyze the care work of Latin American women caregivers, their perception of working conditions and their relationships with the people they care for and their families.	To analyze how the Latin American women caregivers in Mallorca are misrecognized and colonialized	To describe access and utilisation of social and healthcare services by undocumented Latin American women working and living in rural and urban areas, and the barriers these women may face	To gain understanding of daily life participation in occupation experiences of Latin American women living in vulnerable situations, and to identify mediators of health and wellbeing in order to propose collaborative actions intended to promote health
Methodologies	Qualitative approach underpinned by Social Critical Paradigm	Mix-methods multisite study. Qualitative approach underpinned by Social Critical Paradigm	Qualitative inquiry underpinned by Social Critical Paradigm	Participatory health research study underpinned by Social Critical Paradigm

**Table 2 ijerph-17-08278-t002:** Main Self-Reported Health Consequences of Intimate Labour.

Health Problems	Example
Pain	“In my first job, I was really sick because of back pain; I could no longer move very well because I had to move the lady from the bedroom to the bathroom [and] from the bathroom to the living room. I have always been skinny, so I really had a hard time.” (Study 1, Ruth)
Fatigue	“A lot [of fatigue] because, as a live-in maid, you have to work like sixteen hours a day; although they give you a one-hour break, when there are small children, you cannot rest unless when the children are resting, and as much as a child is [calm]—he does not… he does not rest—he is always asking for attention, so it is difficult. I ended up exhausted because I was used to more, more movement always outside. I lasted a year and a half, but I left! Phew! More than exhausted.” (Study 1, Alba) “I feel exhausted all the time, always… (…) but before I start working, I am so... exhausted, extremely exhausted... and this, this is what has mainly affected my health; my sight too—I have lost my sight... (…) the bones, pain in the bones... general tiredness.” (Study 1, Rocío)
Sleep deprivation	“I already feel that I am bleeding from (my nose...) and my health is [deteriorating] because it is 10 times; I get up every night 10 times. Every hour the lady goes to the bathroom and does not pee in her diaper!” (Study 4, Olivia)
Skin irritations	“When I use a lot of detergent and such, as they say, [I have to wear] latex gloves with a lot of dust, which irritates my hands a lot, and I only had that since last year, mid-year. And my whole hand starts to itch, it started to itch, and it was so red and my skin became so stiff (…) the itching started in my hands, reached the elbow and now has spread to the thighs and buttocks.” (Study 1, Ruth)
Poor nutrition	“I think that my diet here is very bad because I often go to work at nine, and until I get home, I don’t eat. And there are times when I get home at five, four-thirty, and I spend the entire day without eating.” (Study 1, Madrid, M. Carmen) “Coming home and eating late affected my health considerably, so that had the strongest effect on [me] (…) yes, I could not even eat; that is, I could not eat quietly. I could not have a quiet breakfast; I was always looking at the time, and I was always on a rush.” (Study 4, Maria)
Anxiety and Depression	“I often feel depressed, which I have never felt before.” (Study 1, Rocío) “I had never known what it was like to be depressed and here... I feel like that so often.” (Study 3, María)
Stress	“I’m a little stressed from being locked up there.” (Study 1, Vanesa)
Frustration	“you look for it [better jobs] (…) [because] you would like to move on, but you know you cannot (…) it is frustrating.” (Study 1, Alba)
Isolation	“I stopped working with the grandmother because I felt that my confinement was over… What was difficult for me here was the schedule; I felt like a prisoner in the house; it destroyed me…” (Study 2, Mariana) “It was difficult; it is not easy to get used to it, that is—being in this situation —being in a house every day without being able to go out, to have a day off per week, it was hard on me because... I am young (…) Well, not being able to go out all day long was terrible, that is, for me going to buy bread was like the ultimate joy of being able to go out and get some sun and fresh air.” (Study 1, Carolina)

**Table 3 ijerph-17-08278-t003:** Coping Strategies Endured by Latin American Immigrant Women.

Strategies	Quotes
Accepting	“This is what it is. In other words, I knew what I was coming to; I knew that I was not going to work in an office, that I was not going to work in a store, no... I knew that I was coming to clean or whatever! In other words, to work in whatever but within limits… “(Study 2, Mónica) “I... that is what I [have] come to......Less than a whore! (laughs)...” (Study 3, Rosaura)
Refusing certain jobs	“…and I can’t do a lot of caregiving work. I have to look at what kind of work. They offer me a job, but I can’t accept it because of my health.” (Study 3, Rosalba) “There are people who accept this situation. I don’t; that is exploitation, terrible, horrible, and humiliating (...) that is why those cases of humiliation and exploitation happen, because people keep quiet and put up with it; what you have to do is not put up with it and not accept it. I go to whoever they are going to hire like that in those conditions. I am not hired for that reason; do not even think of making me work as a mule; I am black but not a slave, my friend” (Study 4, Elena)
Enduring	““…that’s why I came to Spain: to work... and if I have to shut up or bite my tongue, then I bite it. Because I have come to this point.” (Study 3, Rosalba) “and because I have, I don’t have papers, I don’t have the security of... of being able to say: ‘This happens’ – that’s why I sometimes get depressed because I keep everything inside.” (Study 1, Lisethe)
Negotiating with employers	“I told him; when he gave me the new contract, I told him that I was working for him, but that he should give me a day off, because working every day is tiring, so he said yes, that he was going to give me a day off.” (Study 1, Mercedes)
Adapting	“because one has to know how to deal with them.” (Study 1, Ruth) “when they offer you a job, no matter how bad it is, even if you know that they are [exploiting you], you have to thank them” (Study 4, Olivia)
Thinking about quitting or changing jobs	“I was thinking, and my friends also asked me, why aren’t you looking for something external? And then I gained courage, and I said, I will look for external work because it was very bad.” (Study 1, Teófila)
Quitting	“but I thought, I said, and what about my health? It’s worth more than that money, so I told him, I am going to leave, and I left; I’m very sensitive, I was crying a lot (…) and I realized it when I was working: no, work is not more valuable than my health /…/ but one at least has dignity; I have my dignity and I’m not going to be like that either.” (Study 4, María)
Thinking about returning to their countries	“I think you don’t leave because of pride. Because otherwise, one would many times return to one’s country.” (Study 2, Joana)

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
