# Peer review of "The Health Consequences of Neocolonialism for Latin American Immigrant Women Working as Caregivers in Spain: A Multisite Qualitative Analysis"

_ijerph, 2020, doi:10.3390/ijerph17218278_

Round 1
Reviewer 1 Report
This is an interesting paper on the effects of neocolonialism in Spain in immigrant Latin American women working in intimate labour.
Although the results of the studies are very indicative of harsh working reality, I would like to see some reference in local socially deprived women working in IL in order to compare results and put things into a better context. Certainly migratory status predisposes to discrimination, but more essentially the financial need and dependence play vital role in working exploitation and mistreatment.
Moreover, it would be useful to see some descriptive characteristics of the participating women such as family status, length of stay in Spain, educational level, presence of illness or disability, live-in or leave out status and reason for migration.
It would be useful to have an idea of the prerequisites for a migrant to get the legal status of stay and work in Spain.
In other European countries, agencies exist that arrange job interviews for IL for a small fee and this is in many cases a way to express dissatisfaction, report mistreatment from the employer or any other difficulty. Is something similar available in Spain?
Did the women in the sample complete the SF-36 questionnaire?
Finally in the Discussion, one should mention that IL is an inherently physically demanding job as is for people working in old people care homes, physiotherapists etc.
Some minor English check is required.
Author Response
Response to Reviewer 1 Comments
This is an interesting paper on the effects of neocolonialism in Spain in immigrant Latin American women working in intimate labour.
Thank you very much for your review and suggestions to improve the manuscript
Point 1. Although the results of the studies are very indicative of harsh working reality, I would like to see some reference in local socially deprived women working in IL in order to compare results and put things into a better context. Certainly migratory status predisposes to discrimination, but more essentially the financial need and dependence play vital role in working exploitation and mistreatment.
Response 1. Thank you for your comment. It is it is beyond the scope of this study to compare the realities of immigrant workers and local workers. We position our study within a decolonial framework with the aim of exposing the operating mechanisms that situate Latin American women in this sector, which are broadly accepted in the social sciences, and the self-reported consequences in health and wellbeing of such intricated situatedness. Comparing such mechanisms or their implications to those operating for locals would entail a new research.
Point 2. Moreover, it would be useful to see some descriptive characteristics of the participating women such as family status, length of stay in Spain, educational level, presence of illness or disability, live-in or leave out status and reason for migration.
Response 2. We had already included available data of participants in table 1. In response to your comments, we have now added an explanation into the methodology section (page 5) about some of your concerns: “Participants’ length of stay in Spain ranged from one year to 14 years”, and “intimate labour for many participants entailed cycles of live-in and live-out workers.”. Some of the variables were not recruited in the 4 studies, so now this limitation is explained under the discussion section on page 17: “some variables (such as educational level, family status, presence of previous illnesses, and reason for migration) were not registered”; and, about the presence of disability, it seems obvious to us that women should be healthy enough to be able to conduct the required work.
Point 3. It would be useful to have an idea of the prerequisites for a migrant to get the legal status of stay and work in Spain.
Response 3. We explain below the prerequisites:
There are different types of residency cards for people from a non-EU country to become a legal resident and worker: 1) temporary residency, which will grant the possibility to legally live and work in the country for 1 or 2 years. Once that period finished, and provided that the people still meet the initial requirements, people can renew their temporary residence permit for an additional year or two; 2) permanent residency: after 5 years of living uninterruptedly in Spain with a permit or visa, one won’t have to renew it anymore because then one will be eligible to apply for the permanent residency (3), a permit that will allow you to live and work in Spain indefinitely. 4) The arraigo (also called roots procedure) is the process that allows any non-European citizen who has been living in Spain illegally for a certain amount of time to obtain the residency in the country and regularize their situation. Thus, this residence authorization due to exceptional circumstances will allow one to live in the Spanish territory for one year; with the possibility to renew it and modify it to a regular residence permit. Through the roots procedure, one is allowed to legally live and work in the Spanish territory both for a company or as a self-employed individual in any Spanish region. And that thanks to the regulation included on the Organic Law 4/2000 and the Royal Decree 557/2011.
We consider that adding such information to the manuscript it is not necessary, since it would not add specific information which help to better understand the results.
Point 4. In other European countries, agencies exist that arrange job interviews for IL for a small fee and this is in many cases a way to express dissatisfaction, report mistreatment from the employer or any other difficulty. Is something similar available in Spain?
Response 3. The widespread mechanisms of getting a job in the domestic service (IL), which is by word of mouth and informal networks. A few programs were funded by Spanish Government at the onset of the migration flow from Latin America, and mostly NGO would intermediate between place of origin and destination, including arranging job interviews. Nevertheless, these are really exceptions, which explains why the IL forms part of the irregular market and black economy, without formal arrangements and subsequent loss of rights for the workers, as this study shows.
Point 4. Did the women in the sample complete the SF-36 questionnaire?
Response 4. No, only participants from original study 1 completed the SF-36 questionnaire, since the results of this quantitative study are included to build the case for the research in reference number 38. It was not relevant to use the SF-36 to achieve our study aim: the objective of this study was to explain how the colonial logic mediate Latin American women' experiences working in IL in Spain, and the effects of such occupation for their health and wellbeing. Thus, qualitative methodology is the suitable approach to achieve this study aim.
Point 5. Finally in the Discussion, one should mention that IL is an inherently physically demanding job as is for people working in old people care homes, physiotherapists etc.
Response 5. Thanks. In this case we don’t agree that it's any more physical than the other two dimensions of health. In fact, one of the main results of the study is that the physical, psychological and social demands are enormous. For example, the reality of residential workers –with regulated working permit, employment conditions, etc- is immensely different.
Point 6. Some minor English check is required.
Response 6. Thank you for your comment. English proofreading was performed by a native speaker, as you can see now in the version of the manuscript small language editings written in red.

Reviewer 2 Report
This is an interesting and well theorised study that explores the role of colonial ideology/logic as it pervades Latin American migrant women's experiences of working in intimate caring/domestic roles in Spain. The study is described as a multisite qualitative analysis using secondary data across four previously conducted studies. The study provides a reasonably well justified and contextualised analysis of the secondary data. It will be of interest to the readers of the journal, but there are some improvements that can be made to the paper prior to publication.
Methods: The main area for improvement is to provide a more detailed and critical discussion of the methodology and research design. Although multisite qualitative studies are cited (e..g Jenkins), it would help if the authors would go into a little detail as to the strengths and weaknesses of this approach, the rationale for taking this approach, and the use of seconary data analysis for this. The approach has some similarities with other forms of methodology that synthesise qualitative studies (e.g. meta-ethography), so it would help if the authors gave a more robust defence of this approach, contextualising it within the available options.
Table 1 - perhaps needs to mention underpinning methodology, if this was noted. Differences in methodology can lead to quite different quality of data.
Data analysis - the documentation of this is fairly transparent, but it would be good to know how the data was analysed in the four studies - are they using a similar or complementary approach? Is the approach used here a recognised (and referenced) framework of secondary analysis for qualitative data?
Results: The categories seem appropriate and well justified, but one section - 3.2.2 (postive experiences) did not seem to fit in well with the category of 'working conditions based on colonial relationships). It may be best to either work this section around the notion of colonial logic, or explain that there were some deviant cases that did not fit the prevailing analysis but that this data can be explained in a number of ways.
Table 2 - referred to as 'health consequences', but I think 'self-reported health consequences' would be more apposite.
Section 3.3. - forms of resistance - this is a very interesting section as it explores more the agentic dimensions of migrant women's experiences, but the table reveals some imbalance in the use of the studies.. it could be that the resistance was more a feature of some of the studies. For example, the illustrations reveal study 1 (used four times), study 2 (once), study 3 (three times), and study 4 (eight times - mostly one person). I would try to balance this out more or have a discussion about the socio-cultural/economic mechanics of this resistance.
References: the numbering of the references is wrong - 10 follows 1, and other lower numbers are jumbled up in the list. This will need correcting alongside corresponding citations.
Introduction: 'healthy immigrant effect' - needs defining.
The paper would benefit from a thorough proof read as there were a number of simple typographical and stylistic errors.
Author Response
Response to Reviewer 2 Comments
This is an interesting and well theorised study that explores the role of colonial ideology/logic as it pervades Latin American migrant women's experiences of working in intimate caring/domestic roles in Spain. The study is described as a multisite qualitative analysis using secondary data across four previously conducted studies. The study provides a reasonably well justified and contextualised analysis of the secondary data. It will be of interest to the readers of the journal, but there are some improvements that can be made to the paper prior to publication.
Thank you very much for your review and suggestions to improve the manuscript
Point 1. Methods: The main area for improvement is to provide a more detailed and critical discussion of the methodology and research design. Although multisite qualitative studies are cited (e..g Jenkins), it would help if the authors would go into a little detail as to the strengths and weaknesses of this approach, the rationale for taking this approach, and the use of seconary data analysis for this. The approach has some similarities with other forms of methodology that synthesise qualitative studies (e.g. meta-ethography), so it would help if the authors gave a more robust defence of this approach, contextualising it within the available options.
Response 1: We agree that diverse efforts have been made to overcome criticisms to the limitations of qualitative research findings to being too specific, limiting its usability in practice, policy making or in different realities to the original study. Multisite qualitative studies and qualitative meta-synthesis are some examples of such efforts. Whereas qualitative meta-synthesis is an interpretation of the findings of a group of studies, and not a secondary analysis of their data, multisite qualitative studies allow to revisit and analyze the raw data of a set of studies to answer a new research question and are suited to increase transferability and trustworthiness of contextually-relevant findings.
Sandelowski, M., & Barroso, J. (2007). Handbook for synthesizing qualitative research. New York, NY:
Springer
Zimmer, L. (2006). Qualitative meta-synthesis: A question of dialoguing with texts. Journal of Advanced
Nursing, 53, 311-318.
Thus, methodology section has been expanded including a stronger rationale for the use of multisite qualitative studies, as you can read now on page 4:
“Multisite qualitative studies allow to revisit and analyze the raw data of a set of studies to answer a new research question and are suited to increase transferability and trustworthiness of contextually-relevant findings52. We used Jenkins52 multisite qualitative analysis to generate new understandings of immigrants’ circumstances in each site, transcending the specificities of each location while keeping the relevant contextual information allowing to identify the health consequences of performing IL and working in a country operating under a neocolonial logic for Latin American immigrant women. The original studies could be re-analysed due to their shared critical onto-epistemological position that guided their theoretical framework and methodologies, and the substantive interest in participants’ health and wellbeing“
Point 2. Table 1 - perhaps needs to mention underpinning methodology, if this was noted. Differences in methodology can lead to quite different quality of data.
Point 3. Data analysis - the documentation of this is fairly transparent, but it would be good to know how the data was analysed in the four studies - are they using a similar or complementary approach? Is the approach used here a recognised (and referenced) framework of secondary analysis for qualitative data?
Response to points 2 and 3: Thank you for your comment. Table 1 has been expanded including the required information. We added a new file in table 1 detailing the original analysis approaches, as you can see now on page 4.
|
|
S1 |
S2 |
S3 |
S4 |
|
Methodologies |
Qualitative approach underpinned by Social Critical Paradigm |
Mix-methods multisite study. Qualitative approach underpinned by Social Critical Paradigm |
Qualitative inquiry underpinned by Social Critical Paradigm |
Participatory health research study underpinned by Social Critical Paradigm |
Point 4. Results: The categories seem appropriate and well justified, but one section - 3.2.2 (postive experiences) did not seem to fit in well with the category of 'working conditions based on colonial relationships). It may be best to either work this section around the notion of colonial logic, or explain that there were some deviant cases that did not fit the prevailing analysis but that this data can be explained in a number of ways.
Thank you for your comment. We have renamed the sub-section to “Perceived positive elements buffering IL health consequences”, as you can see now on page 11. Participants account of those experiences are repeatedly named as positive or good. We believe it is relevant to maintain such denomination, acknowledging participants’ views. Nevertheless, during this subsection, the notion of colonial logic is addressed by explaining how these perceived positive elements buffer but do not change nor eliminate, the colonial logic underpinning this occupation, the subsequent dynamics and the implications for health and wellbeing. A clarificatory sentences has been added to reinforce such idea, as you can see now on page 11 and below:
These positive features buffered the negative impact of being an immigrant women performing IL but do not change the intrinsic colonial dynamics that characterised IL for participants.
Point 5. Table 2 - referred to as 'health consequences', but I think 'self-reported health consequences' would be more apposite.
Response 5. Thank you for your comment. Table 2 has been renamed as suggested, as you can see now on page 10
Table 2. Main Self-Reported Health Consequences of Intimate Labour
Point 6. Section 3.3. - forms of resistance - this is a very interesting section as it explores more the agentic dimensions of migrant women's experiences, but the table reveals some imbalance in the use of the studies.. it could be that the resistance was more a feature of some of the studies. For example, the illustrations reveal study 1 (used four times), study 2 (once), study 3 (three times), and study 4 (eight times - mostly one person). I would try to balance this out more or have a discussion about the socio-cultural/economic mechanics of this resistance.
Thank you for your remark, we agree that study 4 was overrepresented in this section of results, so we tried to balance the illustrating identified quotations in all of the 4 studies. Thus, we removed some quotes of study 4 and added a new one of study 2, as you can see now on page 14 .
Point 7. References: the numbering of the references is wrong - 10 follows 1, and other lower numbers are jumbled up in the list. This will need correcting alongside corresponding citations.
Response 7. Thank you for your indication, there was a mistake between draft versions and references 2-9 were missing. This issue has been corrected as you can see in the new version of the references list
Point 8. Introduction: 'healthy immigrant effect' - needs defining.
Response 8. Thanks. We added a definition as you can read now on page 3 and below:
“The loss of the healthy immigrant effect, according to that migrants may have a better health status than the natives -because those in the best health are who migrate-, may also be related to the impact of the 2008 economic crisis”
Point 9. The paper would benefit from a thorough proof read as there were a number of simple typographical and stylistic errors.
Response 9. Thank you for your comment. English proofreading was performed by a native speaker, as you can see now small language edits in red in the new version of the manuscript

Reviewer 3 Report
The topic is of interest.
I have some concerns.
Introduction and concepts
It would have been of interest to describe the introduction to the topic comparing the situation with other cultures and countries in this area "caregivers". Other countries seem to have a similar situation. For example in Germany also immigrants work as caregivers and it seems that the working conditions are also precarious. Here the immigrants that are working as caregivers are coming from Eastern European countries and Rusia etc. So, it will be enriching to work out what makes the difference between Spain and these other countries.
The justification of the conceptual framework and the detailed concepts of the framework are not described with sufficient detail. Therefore, it remains unclear why a certain concept was selected and which is the advantage when using this concept and not another one. This should be worked out.
1.2. Migration and Health
There is an interesting body of literature on acculturation and cultural hassles of immigrants from scholar such as Safdar. Results from these research studies would help to detail and deepen this conceptual part.
2. Methodology
Sample
It would be helpful to have some more information about the participants: such as educational level, in-group contact etc. to understand better the characteristics of the sample.
The sample comes from different Latin American cultures with different cultural nuances. I wonder wether this could have an influence. This aspect should be mentioned and debated later also under the "Limitations" section.
Procedure and data collection
The timeframe of six years is long. Situations of immigrants and the host country and host society can change over six years. Several hidden variables can influence this panorama. It is unclear how this aspect was controlled.
2.3. Data generation and interview questions
It remains still unclear which conceptual areas were selected and how and which questions were developed.
2.5. Quality criteria
Was the reliability analyzed?
3. Results
Authors mention a mediation. More information on the mediation analysis is needed to be able to replicate the study.
Health consequences of the intimate work: causalities cannot really assured. This should be mentioned.
Translation:
The English translation of the quotes was double checked following the guidelines of the International test Commission or other guidelines?
4.2. Limitations:
"Differences in interview guides..." It remains unclear what exactly the authors want to say here. This should be cleared up in all corresponding manuscript sections.
Future research recommendations are missing.
Author Response
Response to Reviewer 3 Comments
The topic is of interest.
Response. Thank you for your positive comment and all your suggestions to improve the manuscript.
I have some concerns.
Introduction and concepts
Point 1. It would have been of interest to describe the introduction to the topic comparing the situation with other cultures and countries in this area "caregivers". Other countries seem to have a similar situation. For example in Germany also immigrants work as caregivers and it seems that the working conditions are also precarious. Here the immigrants that are working as caregivers are coming from Eastern European countries and Rusia etc. So, it will be enriching to work out what makes the difference between Spain and these other countries.
Response to point 1. Despite we consider that on page 2 we already explained why Spain could be especially attractive to Latin American women, following your suggestion we specify now that is a common pattern in Europe and we added a new reference, as you can see now on page 2 and below:
In the current global economic system, full-time employed women in high-income countries need a substitute for their domestic chores and caregiving activities, usually an immigrant woman13. In Spain, the role of caregiving and house cleaning have been mainly delegated to Latin American women, as they come from former Spanish colonies and share cultural and language backgrounds14. As in all Europe, for caregivers a significant portion of job opportunities is in the informal market, offering precarious employment with no or minimal labour rights and social security7.
Jennifer Rubin, Michael S. Rendall, Lila Rabinovich, et al. Migrant women in the European labour force Current situation and future prospects. RAND Europe, 2008. Available at: https://www.rand.org/pubs/technical_reports/TR591.html
And on page 5:
Such logic prevails in Western countries, where the majority of domestic workers are migrants and/or belonging to ethnic minorities.
Point 2. The justification of the conceptual framework and the detailed concepts of the framework are not described with sufficient detail. Therefore, it remains unclear why a certain concept was selected and which is the advantage when using this concept and not another one. This should be worked out.
Response point 2. We apologize since we are not sure of having totally understood this comment. We consider that the decolonial conceptual framework is explained in point 1.1., and it is justified since both matches with the objective of the study “to explain how the colonial logic mediate Latin American women' experiences working in IL in Spain, and the effects of such occupation for their health and wellbeing”, and coherently guides all the study to expose the current effects of the colonial logic on Latin migrant women IL workers in Spain.
Point 3. 1.2. Migration and Health
There is an interesting body of literature on acculturation and cultural hassles of immigrants from scholar such as Safdar. Results from these research studies would help to detail and deepen this conceptual part.
Response to point 3. Thank you for your comment. We have positioned our research within a decolonial framework and used the according literature to support and elaborate our study. Although the area of acculturation and cultural hassles may explain some of the issues encountered by participants, we believe, as elaborated in our study, that the decolonial framework is better prepared for understanding this phenomenon.
- Methodology
Sample
Point 4. It would be helpful to have some more information about the participants: such as educational level, in-group contact etc. to understand better the characteristics of the sample.
Response 4: Thank you for your comment. We have included available data of participants in table 1 to provide a better description, Nevertheless, some variables such as educational level were not registered (which has been added to the limitations section, limitation n.4, page 17). We apologize since we do not understand properly the expression “in-group contact”.
Point 5. The sample comes from different Latin American cultures with different cultural nuances. I wonder wether this could have an influence. This aspect should be mentioned and debated later also under the "Limitations" section.
Response 5. Thank you for your comment. Despite we already explained under the former version of limitations that “Diversity in participants’ nationalities helped us to identify the shared experiences of Latin American women working in IL, which is a Spanish social construct that supports a shared social imaginary which is not directly related to any specific country”. We broaden now the limitations on page 17, as you can read below:
“It should be taken into account that in this multisite qualitative study: /…/ 2) participants’ diverse origin countries and associated cultures may have nuanced some of their experiences, /…/ Nevertheless, we consider that: /…/ 2) diversity in participants’ nationalities helped us to identify the shared experiences of Latin American women working in IL, which is a Spanish social construct that supports a shared social imaginary which is not directly related to any specific country.”
Procedure and data collection
Point 6. The timeframe of six years is long. Situations of immigrants and the host country and host society can change over six years. Several hidden variables can influence this panorama. It is unclear how this aspect was controlled.
Response 6. Thank you for your comment. We have addressed this issue under the discussion section, as you can read now on page 17 and below:
It should be taken into account that in this multisite qualitative study: 1) the original studies took place 6 years timeframe (2011-15), /…/ Nevertheless, we consider that: 1) all interviews were conducted after the 2008 crisis began, in the same context of increasing vulnerability for immigrant workers.
Point 7. 2.3. Data generation and interview questions
It remains still unclear which conceptual areas were selected and how and which questions were developed.
We identified such parts of the transcriptions of original studies relating to our research question (which generated the objective) “how were women' experiences working in IL in Spain, and the effects of such occupation for their health and wellbeing?”. For example, we did not included references in the transcriptions about migration stories or their family relations.
Point 8. 2.5. Quality criteria
Was the reliability analyzed?
Response 6. Thanks. We reflect into this under limitations section, such you can now read on page 17 and below: “Different measures were taken in this study to guarantee reliability of qualitative research, such as triangulation, a detailed context of the study, and use of literal citations”
Point 9. 3. Results
Authors mention a mediation. More information on the mediation analysis is needed to be able to replicate the study.
In the former version of the manuscript we wrote that trained cultural mediators conducted some of the interviews. One of the interviewers in study 3 was an Arabic women, who also worked as cultural mediator, but in this study acted just as a research assistant. So we decided to include her as research assistant instead of cultural mediator, to avoid confusion, so we removed this expression of the paragraph on data generation (page 5).
Point 10. Health consequences of the intimate work: causalities cannot really assured. This should be mentioned.
Reponse 10. We agree. We added in limitations and strengths (page 17):
“Although the results are not able to assure causality, they can be used to theorize about the health consequences of IL in Spain and similar settings from women’s perspective..”
Translation:
Point 11. The English translation of the quotes was double checked following the guidelines of the International test Commission or other guidelines?
Response 11. No, we did not follow any specific guidelines. Sorry, it is the first time we read about it, despite our time has a long experience in publishing qualitative studies.
4.2. Limitations:
Point 12. "Differences in interview guides..." It remains unclear what exactly the authors want to say here. This should be cleared up in all corresponding manuscript sections.
Response point 12. We apologize if this expression was not well understood. We were referring to the fact that, since the secondary analysis includes 4 different studies, each study had its own interview guide; nevertheless the studies share similar objectives (as you can see now in table 1) and then asked similar questions related with the migration process, their integration in Spain (including the job market) and access to resources. We decided to remove this limitation since it could be confusing for the reader and we already highlighted the similarities between studies.
Point 14. Future research recommendations are missing.
Response to poing 14. We are afraid that we don’t agree in this point, since already in page 17 we included a sentence about that: “and new research to explore the specific situation of this vulnerable group in relation to the Covid-19 pandemic.”

Reviewer 4 Report
Thank you for this well written manuscript. The focus on using a decolonial framework to explore women’s experiences and related health consequences is a strength of this manuscript. My suggestions are minor and outlined below:
- For ethical approval, how was ethics approval for secondary analysis obtained? Was consent given in the primary studies for secondary analysis, etc.?
- It would be helpful to identify the main objective of each of the individual studies in Table 1 as it would give the reader a better understanding of the type of data collected for the primary study.
- In table 3 the strategy “resisting” seems to be more about “enduring” based on reading the quotes provided. I suggest changing the quotes or the strategy title.
- Themes2 & sub-theme 3.2.2 seem to have similar thematic titles. Maybe a more specific label for 3.2.2 would bring more clarity to the reader on how this fits as a sub-theme to the larger heading (theme).
Author Response
Response to Reviewer 4 Comments
Thank you for this well written manuscript. The focus on using a decolonial framework to explore women’s experiences and related health consequences is a strength of this manuscript. My suggestions are minor and outlined below:
Thanks, we really appreciate your comments and suggestions to improve the manuscript.
- Point 1. For ethical approval, how was ethics approval for secondary analysis obtained? Was consent given in the primary studies for secondary analysis, etc.?
- Response 1: Thank you for your comment. This issue has been clarified. Consent for data analysis with research purposes was given in the original studies, so asking for a new consent for the secondary analysis was not needed. We added now on Page 6:
- “Participation in each original study was voluntary, and a consent form authorizing the use of the data with research purposes was signed by all participants”
- Point 2. It would be helpful to identify the main objective of each of the individual studies in Table 1 as it would give the reader a better understanding of the type of data collected for the primary study.
- Response 2: Table 1 has been expanded including the required information, as you can see now on page 4 and below:
|
Table 1. Characteristics of the sample and original studies |
||||
|
|
Study 1 |
Study 2 |
Study 3 |
Study 4 |
|
Objectives |
To analyze the care work of Latin American women caregivers, their perception of working conditions and their relationships with the people they care for and their families. |
To analyze how the Latin American women caregivers in Mallorca are misrecognized and colonialized |
To describe access and utilisation of social and healthcare services by undocumented Latin American women working and living in rural and urban areas, and the barriers these women may face
|
To gain understanding of daily life participation in occupation experiences of Latin American women living in vulnerable situations, and to identify mediators of health and wellbeing in order to propose collaborative actions intended to promote health |
- Point 3. In table 3 the strategy “resisting” seems to be more about “enduring” based on reading the quotes provided. I suggest changing the quotes or the strategy title.
- Response 3: thank you for your suggestion. The strategy title has been changed as suggested.
- Point 4. Themes2 & sub-theme 3.2.2 seem to have similar thematic titles. Maybe a more specific label for 3.2.2 would bring more clarity to the reader on how this fits as a sub-theme to the larger heading (theme).
Response to point 4. Thanks for this remark, we acknowledged that it was a mistake. We corrected the second 3.2.2. by 3.2.3. Now on page 11

Round 2
Reviewer 2 Report
The authors have provided a very good response to my suggestions and as a result the paper is much improved, particularly regarding the soundness, transparency and rigour of the methodology.
Just one note: the references 2-9 have been added, but they also show up at later points during the reference list (I suspect as a result of stylistic/numbering issues in the reference list).The authors will need to take out the duplication.
Author Response
We really appreciate the positive feedback of the reviewer 2.
We also acknowledge the point about the repeated references; as a result, now we uploaded a new version of the manuscript with the duplicated references removed.
Thank you very much